# Differentiable Design With Dynamic Programming

## Kelly O. Marshall, Minsu Cho, Chinmay Hegde

New York University Tandon
km3888@nyu.edu, mc8065@nyu.edu, chinmay.h@nyu.edu

## Abstract

We consider problems in learning-based design subject to constraints specified in the form of Dynamic Programming (DP). Recent work from Mensch and Blondel (2018) proposes the use of a differentiable DP operator, therefore enabling DP constraints to be used in conjunction with gradient-based learning. In this paper, we introduce a differentiable technique called *soft-DP* that can be used to solve target-matching problems using gradient-based methods. Our technique also enables backpropagating "through" DP solutions that obey a piecewise-linear structure. To validate our approach, we report results from three showcase applications – game design, histogram approximation, and materials design – where our approach improves over data-heavy alternatives.

## Motivation

Gradient-based learning has underpinned many of the considerable advances made in artificial intelligence and machine learning over the last decade. Correspondingly, significant attention has been devoted to creating *differentiable* modules that solve optimization problems. These are commonly used either as loss functions or as "layers" within larger models (Amos and Kolter 2017; Wang et al. 2019; Lample et al. 2016; Mensch and Blondel 2018).

In this paper, we focus on Dynamic Programming (DP), a fundamental algorithmic technique for finding efficient solutions to combinatorial optimization problems (Bellman 1957). We develop a differentiable DP approximation, that we call *soft-DP*, to design algorithms that can solve design problems involving dynamic programming using gradient updates. In some design problems, finding a solution requires not just minimizing or maximizing DP-defined attributes, but having them match specific target values. We show how to adapt soft-DP for target-matching, and also show how soft-DP enables approximating gradients of *solutions* to DP problems that exhibit piecewise output structure.

Our numerical experiments confirm the effectiveness of soft-DP in three showcase applications: (i) game level design, (ii) histogram-approximation, and (iii) material microstructure reconstruction, and improvement over data-heavy methods that involve training neural network surrogates for gradient computation.

## Preliminaries

Let us set up some notation. Denote $[n] = \{1, 2, \ldots, n\}$ as an index set. Let $\Delta^d$ be the probability simplex of dimension $d$. For ordered graphs, we use $\mathcal{P}_i$ and $\mathcal{C}_i$ to denote the set of parent nodes and child nodes for node $v_i$.

### Differentiable Dynamic Programming

Dynamic Programming (DP) is a family of algorithms that can be used to efficiently solve many combinatorial optimization problems. DP algorithms exploit the existence of overlapping sub-problems by storing the solutions to these sub-problems for later use (Bellman 1957). All DP approaches can be formulated as operations on a Directed Acyclic Graph (DAG) with topologically ordered nodes $v_1, .., v_n$ and edge weights $\theta \in \mathbb{R}^{N \times N}$. The solution to the original problem then reduces to finding the optimal path on this graph that maximizes (or minimizes) the sum of edge weights. This can be done through the following recursion:

$$v_1(\theta) \triangleq 0 \tag{1}$$
$$\forall i \in \{2, ..., N\} : v_i(\theta) \triangleq \max_{j \in \mathcal{P}_i} \theta_{ij} + v_j(\theta) \tag{2}$$

The final output $V(\theta) \triangleq v_N(\theta)$ is the optimal cost and can be shown to be the same as a result obtained by performing an exhaustive search over all possible paths. By storing the maximizing index for each step, the *solution* of the dynamic program can then be reconstructed through backtracking from the final node of the graph, giving the optimal path.

The mapping $V(\theta)$: $\mathbb{R}^{N \times N} \to \mathbb{R}$ is non-differentiable wherever the solution is not unique; however, previous work (Mensch and Blondel 2018) has devised a method for computing a differentiable approximation by adding a strongly-convex regularizer to each of the recursive updates. For an arbitrary regularizer $\Omega$, they leverage the *smoothed max* operator $\max_{\Omega}(x)$ introduced by (Nesterov 2005), defined as follows:

$$\max_{\Omega}(\mathbf{x}) = \max_{q \in \Delta^{|\mathbf{x}|}} \langle \mathbf{q}, \mathbf{x} \rangle - \Omega(\mathbf{q}) \tag{3}$$

Unlike the regular max operator, this smoothed version is continuously differentiable everywhere. By substituting it into the recursive updates in Equation 2, we can define a smooth DP approximator $V_{\Omega}(\theta)$ with gradients $\nabla_{\theta} V_{\Omega}(\theta)$ that are guaranteed to exist. The approximation is also prin-

cipled in the sense that $\lim_{\alpha \to 0} V_{\alpha\Omega}(\theta) = V(\theta)$ for any choice of $\Omega$.

In this paper we focus on using the negative entropy regularizer $\Omega$, which yields the well-known softmax function (Bridle 1990). Note that for DP problems that call for minimization over arguments, it is straightforward to define an analogous smoothed min operator: $\min_\Omega(\mathbf{x}) \triangleq -\max_\Omega(-x)$.

To allow the computation of input gradients $\nabla_\theta V_\Omega(x)$ in the backward pass, the maximizing $q$ vectors from (3) must be stored in the forward pass, with $q_i$ denoting the maximizing argument used to compute $v_i$. Together, these define transition probabilities for a random walk on the input graph, starting from the final node and working backwards. The backward pass consists of backtracking through the DAG to compute the marginal probability of each node and edge being visited during this random walk. The marginal probability of each edge being visited is equivalent to the gradient of $V_\Omega(\theta)$ with respect to that edge.

## Differentiable $k$-Histogram Approximation

Given a signal $\mathbf{x} \in \mathbb{R}^n$, a k-histogram approximation of $\mathbf{x}$ is obtained via piecewise constant regression problem, i.e., the signal $\mathbf{x}$ is approximated by a set of $k$ constant segments. Jagadish et al. (1998) leverages a dynamic programming approach to solve the $k$-histogram approximation with $O(kn^2)$ time complexity.

Denote the function $h : \mathbb{R}^n \to \mathbb{R}^n$ as the map from a given time-series signal to its $k$-histogram approximation via dynamic programming approach. Cho et al. (2021) estimates an analytic, closed-form form of the Jacobian of $h$ by leveraging the partition of $[n]$ produced from DP. Formally, let $\Pi = \{\mathbf{B}_1, \ldots, \mathbf{B}_k\}$ denote any partition of $[n]$. The dynamic programming approach equivalently solves the following optimization problem:

$$\min_{\mathbf{B}_1, \ldots, \mathbf{B}_k} \sum_{i=1}^{k} \sum_{j \in \mathbf{B}_i} (x_j - \frac{1}{|\mathbf{B}_i|} \sum_{l \in \mathbf{B}_i} x_l)^2$$

Cho et al. (2021) showed that the (weak) Jacobian $\partial h / \partial \mathbf{x}$ forms the block-diagonal matrix $\mathbf{J} \in \mathbb{R}^{n \times n}$:

$$\mathbf{J} = \begin{bmatrix} \mathbf{J}_1 & \cdots & \mathbf{0} \\ \vdots & \ddots & \vdots \\ \mathbf{0} & \cdots & \mathbf{J}_k \end{bmatrix}$$

where all entries of each block $\mathbf{J}_i \in \mathbb{R}^{|\mathbf{B}_i| \times |\mathbf{B}_i|}$ are equal to the reciprocal of the number of elements in $\mathbf{B}_i$.

## Our Proposed Approach

**Target matching.** We consider design problems where the design variable $\mathbf{x}$ is defined such that some attribute $V(\mathbf{x})$ — which is computed using dynamic programming — has to be equal to a desired value, while potentially satisfying a number of other constraints.

The mapping $x \mapsto V(x)$ is typically non-differentiable, and therefore one we need to resort to alternatives. One approach would be to train a neural network as a surrogate for $V(x)$ and leveraging automatic differentiation. But this, of course, requires an potentially large auxiliary dataset of

$(x, V(x))$ labeled training samples.

Alternatively, we could add a strongly convex regularizer into the DP-objective as in (Mensch and Blondel 2018). However, the drawback is that this approximation introduces *bias* to the optimal cost. Specifically, the outputted cost approximation will be greater than the true value when performing a minimization over arguments (the opposite will be true when performing a maximization). While Mensch and Blondel (2018) are able to bound this error $|V(\mathbf{x}) - V_\Omega(\mathbf{x})|$, their guarantees involve a linear dependence on the number of DP steps needed for the computation. In practice, some dynamic programs (such as in our experiments on Microstructure Design) require a very large number of steps, resulting in *extremely inaccurate output values*. This poses a challenge in design settings where we wish to use the differentiable DP approach of Mensch and Blondel (2018).

Let us illustrate this. Consider the general case in which we would like to update an input $\mathbf{x}$ using gradient descent over the squared error $\mathcal{L}(\mathbf{x}) = \frac{1}{2}(V(\mathbf{x}) - c)^2$. To get a differentiable loss, we first replace $V$ with $V_\Omega$. Using the chain rule, the gradient of this loss with respect to $x$ can be written as:

$$\nabla_x \mathcal{L}(\mathbf{x}) = \frac{\partial \mathcal{L}}{V_\Omega(\mathbf{x})} \nabla_\mathbf{x} V_\Omega(\mathbf{x}),$$
$$= (V_\Omega(\mathbf{x}) - c) \nabla_\mathbf{x} V_\Omega(\mathbf{x}) \quad (4)$$

Here we can see that the inaccuracy of $V_\Omega$ creates problems as the term $(V_\Omega(\mathbf{x}) - c)$ may not be of the same magnitude – or *even the same sign* as $(V(x) - c)$ – preventing $V(x)$ from ever converging to $c$.

We propose the following. we need to use both $V$ and $V_\Omega$ to accurately compute both terms in the right hand side of Equation 4. The key lies in observing that even when we cannot assume that $V_\Omega(\mathbf{x}) \approx V(\mathbf{x})$, we can still use $\nabla V_\Omega$ to update the value of $V$. We can use an exact DP algorithm $V$ to figure out which direction the attribute needs to be shifted, and then by taking the gradient of $V_\Omega$ to determine how to shift $\mathbf{x}$. For each forward pass we then compute both $V(\mathbf{x})$ and $V_\Omega(\mathbf{x})$ and use these to compute the approximate gradient:

$$\nabla_\mathbf{x} \mathcal{L}(\mathbf{x}) \approx (V(\mathbf{x}) - c) \nabla_\mathbf{x} V_\Omega(\mathbf{x}) \quad (5)$$

We call this *debiased soft-DP* and show below that this gradient approximation can effectively be used to optimize the true loss $(V(\mathbf{x}) - c)^2$.

**Gradients of solutions.** In several problems, we are interested in finding inputs to a DP problem which meets a target *output solution*. For instance, consider the histogram approximation function $h : \mathbb{R}^m \to \mathbb{R}^m$ which maps a signal $\mathbf{x} \in \mathbb{R}^m$ to its optimal $k$-histogram approximation. If we use the gradient of $h$ to incrementally update a signal $\mathbf{x}$ to minimize $\mathcal{L}(\mathbf{x}) = ||h(\mathbf{x}) - c||_2^2$, then the optimal cost may be less informative than the actual approximation itself.

To solve the "gradient of solutions" problem, (Mensch and Blondel 2018) propose using the Hessian of $V$. However, this gives the gradient of a solution that is described by probabilities over edges in the DAG corresponding to the DP. This is useful for some problems, but it is not clear how

**Algorithm 1: Forward pass**

**Input**: Edge Weights $\theta \in \mathbb{R}^{N \times N}$
**Parameter**: Regularization strength $\Omega$
**Output**: Approximation of Optimal Cost $\hat{v} \in \mathbb{R}$, Local gradients $Q \in \mathbb{R}^{N \times N}$

1: Initialize $V \in \mathbb{R}^N$
2: Initialize $Q \in \mathbb{R}^{N \times N}$
3: $V_1 \leftarrow 0$
4: **for** i=1,...,N **do**
5:      Initialize Options $\in \mathbb{R}^N$
6:      **for** $j = 1, ..., N$ **do**
7:         **if** $j \in \mathcal{P}_i$ **then**
8:            Options$_j \leftarrow \theta_{ji} + V_j$
9:         **else**
10:           Options$_j \leftarrow \infty$
11:         **end if**
12:      **end for**
13:      $Q_i \leftarrow \text{Min}_\Omega(\text{Options})$
14:      $V_i \leftarrow Q_i \cdot \text{Options}$
15: **end for**
16: **return** $V_N, Q$

---

**Algorithm 2: Backward Pass - Cost Gradient**

**Input**: Local gradients $Q \in \mathbb{R}^{N \times N}$, Downstream gradient $\frac{\partial \mathcal{L}}{\partial v}$
**Output**: $E = \frac{\partial v}{\partial \theta} \in \mathbb{R}^{NxN}$, $\frac{\partial \mathcal{L}}{\partial \theta_{ij}}$

1: Initialize $E \in \mathbb{R}^{N \times N}$ with zero entries
2: Initialize $\overline{E} \in \mathbb{R}^N$ with zero entries
3: $\overline{E}_N \leftarrow 1$
4: **for** i=N-1,...,1 **do**
5:      **for** $j \in \mathcal{C}_i$ **do**
6:         $E_{i,j} = Q_{ji} * \overline{E}_j$
7:      **end for**
8:      $\overline{E}_i = \sum_{j=1}^{N} E_{i,j}$
9: **end for**
10: **return** $E, E^T \frac{\partial \mathcal{L}}{\partial v}$

---

**Algorithm 3: Backward Pass - Reconstruction Gradient**

**Input**: $E \in \mathbb{R}^{N \times N}$
**Parameter**: Block Jacobian sub-gradient function $J(i, j)$
**Output**: $\frac{\partial solution}{\partial \theta}$

1: Initialize $A \in \mathbb{R}^{NxN}$ with zero entries
2: **for** $i = 1, ..., N$ **do**
3:      **for** $j \in \mathcal{C}_i$ **do**
4:         $A \leftarrow A + E_{ij} J(i, j)$
5:      **end for**
6: **end for**
7: **return** A

---

to use this solution gradient to relax the actual output of the solution for problems which require reconstructing an output from the DAG solution. For example, computing the output of $h$ requires averaging over the partitions discovered using DP. This is straightforward when the DP gives a single optimal path but is difficult to define when there are a combinatorial number of paths with non-zero probability.

We show that for a broad class of DP-problems with piecewise solutions, we can directly compute an approximation of the solution's gradient using the original signal $x$ and the cost gradient $E = \nabla_\theta V(\theta)$, without explicitly performing any Hessian calculations. In this work, we focus on the example of piecewise-constant $k$-histogram approximation as a simple case, but the extension to more complicated piecewise-smooth outputs is straightforward.

We start from the observation in (Cho et al. 2020) that for each bucket $\mathbf{B}$ in the optimal approximation, the exact Jacobian of $h$ will have a block of entries equal to $\frac{1}{|\mathbf{B}|^2}$. We can view the cost-gradient of each edge weight $(i, j)$ as the probability of having a bucket spanning the segment which corresponds to the edge and therefore the probability of the Jacobian having a block $J(i, j)$ in this location. We can therefore compute an approximation of the Jacobian $\frac{\partial h(\mathbf{x})}{\partial \mathbf{x}}$ by a weighted sum over the edges of the DAG as denoted in 3. As the regularization is scaled to zero, this Jacobian converges to the blockwise DSA Jacobian: $\lim_{\alpha \to 0} \nabla_\mathbf{x} S_{\alpha\Omega}(\mathbf{x}) = \mathbf{J}$.

Unlike the DSA method, however, this smoothed solution gradient does not fix the partition locations before computing the Jacobian. As a result, this can be used to perturb inputs in a way that allows the updated version to have different partition locations from the previous one. We show in our experiments that this gradient formulation permits much more flexible optimization over inputs when compared to DSA.

## Experiments

### Game Design

We first compare the design capabilities of backward-pass approximation to an optimal forward-pass approximation on a simple game called Monster Trainer, a shortest path problem with unknown cost functions first introduced in (Xu et al. 2020). In that work, Xu et al. (2020) show that Graph Neural Networks (GNNs) are well suited to learn DP solutions due to the inherent network structure within the DP algorithm. While GNNs are differentiable by design, we conduct experiments replacing the GNN to a data-free approximation of Bellman-Ford by leveraging the method in Algorithm 1 and 2. Figure 6a demonstrates that our data-free approach debiased soft-DP approach outperforms using the surrogate GNN models proposed in (Xu et al. 2020). We defer the details of Monster Trainer problem and other additional results to the Appendix.

### Optimal Histogram Differentiation

We consider the following toy design problem. Let $\mathbf{x} \in \mathbb{R}^n$ and $\mathbf{y} \in \mathbb{R}^n$ be an input and a $k$-histogram approximation respectively. We consider observing a noisy target $\hat{\mathbf{y}} = \mathbf{y} + \epsilon$ where $\epsilon \sim N(0, \sigma^2)$. Given the prior that the target signal $\mathbf{y}$ is a k-histogram, to recover $\mathbf{y}$ we solve a following optimization problem:

$$\min_{\mathbf{x}} \|h(\mathbf{x}) - \hat{\mathbf{y}}\|_2^2 \tag{6}$$

and measure the distance between histogram approximation of the designed (optimal) input, $h(\mathbf{x})$, with the piecewise constant target signal $\mathbf{y}$.

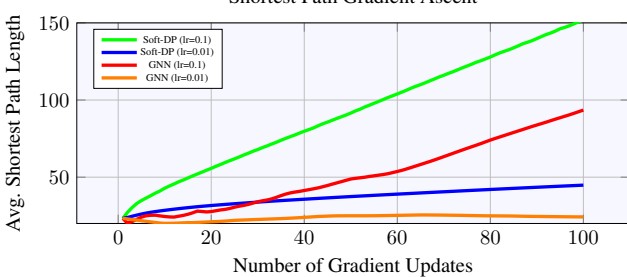

Figure 1: Monster Trainer average shortest path lengths on a batch of 10 randomly sampled levels, updated via gradient ascent. Higher average shortest path length indicate useful gradients.

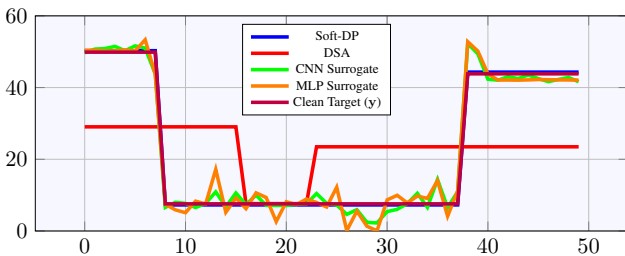

Figure 2: The comparison of $h(\mathbf{x})$ to the clean target. We observe Soft-DP achieves the $h(\mathbf{x})$ the most close to the clean target $\mathbf{y}$. DSA fails to update the input $\mathbf{x}$ that the partition matches to the $\mathbf{y}$. Surrogate models (MLP and CNN) overfit to the noisy target ($\hat{\mathbf{y}}$) failing to achieve close $\ell_2$ distance to $\mathbf{y}$.

We implement the design using several approximations for $h$: Soft-DP, DSA and neural network based surrogate models: multi-layer perceptron (MLP) and convolution neural networks (CNN). We freeze the surrogate models' parameters when solving the optimization problem in Equation 6. Our proposed method using Algorithm 3 is able to update the $h(\mathbf{x})$ close to the clean target while MLP and CNN overfits to the noisy target. We defer the detailed experimental setup and results to the Appendix.

We also compare the Jacobian $\partial h/\partial \mathbf{x}$ between Soft-DP, DSA, MLP, and CNN given histogram inputs $\mathbf{x}$ as shown in Figure 3. The DSA Jacobian is an analytical form of the (weak) Jacobian given the histogram approximation, while Soft-DP adds the probabilistic aspect so that neighboring block matrices are smoothed out if the truth signal partitions are close in terms of the value. On the other hand, we observe that MLP and CNN exhibit noisy Jacobians.

**Microstructure Design**

We now consider a materials design problem from photovoltaics. An important attribute in designing two-phase composite photovoltaic materials (at the microstructural scale) is conductivity, which involves calculating shortest paths from given pixel locations to either the top or bottom of the material volume through the given domain. Motivated by this, we train an generative model, specifically InvNet models (Joshi et al. 2020) (we defer details of InvNet constructions to Ap-

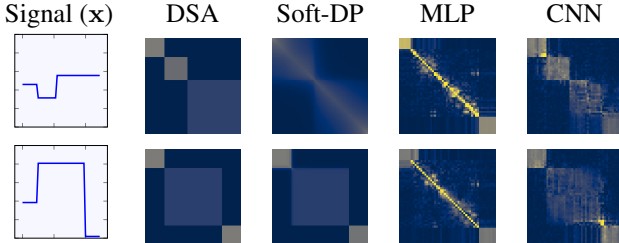

Figure 3: Jacobian comparisons on various histogram approximation methodologies. Each row visualize the Jacobian of DSA, Soft-DP, MLP, and CNN given the signal in the first column. DSA solves the exact Jacobian $\partial h/\mathbf{x}$. We observe that Soft-DP smooths out the neighboring block diagonal matrix if the values in the two given partitions are close to each other (first row). On the other hand, the second row shows no smoothing effect on Soft-DP since the difference in neighboring partitions in $\mathbf{x}$ are large.

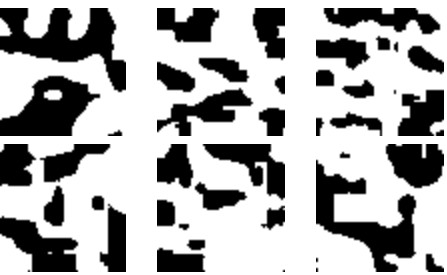

Figure 4: Generated Microstructures

pendix). We use a dataset of 2-phase microstructures and selected an invariance function $f$ defined as the shortest-path from the top to the bottom of the image under some pairwise distance function $d$, which measures the difference between adjacent pixel values. Note that in the following definition, we ensure that $p_1$ is in the first row of the array, $p_2$ is in the last row, and that $\forall i : p_i, p_{i+1}$ are adjacent to one another $SP(\mathcal{I}) = \min_{p_1,p_2...,p_h} \sum_{i=1}^{h-1} d(\mathcal{I}_{p_i}, \mathcal{I}_{p_{i+1}})$. To ensure that identical pixels would have a distance of 1 and that the distance increases rapidly as the values diverged, we choose our distance function as $d(x_1, x_2) = \exp((x_1 - x_2)^2)$.

We define the invariance function via Algorithm 1 and 2. Each microstructure image is of size $64 \times 64$, making for a total of $2^{12}$ DP updates. Due to the high number of DP updates needed to compute $SP_\Omega$, we found it necessary to use our proposed *debiased soft-DP* approach to ensure accurate gradients. This setting gives us an opportunity to test whether the soft-DP gradients can be used to train deep neural networks in addition to the simple input updates we've seen previously.

Our experiments show that the shortest-path relaxation is effective in reducing the invariance cost $(c - SP(x))^2$, meaning that the generator is able to learn to output samples with specified shortest-path values. Increasing the penalty coefficient for the DP-based loss further ensures that the generator obeys the control input as shown in Figure 4. We defer additional experimental results to the Appendix. Our code can be found here.

## Acknowledgements

The authors were supported in part by the NSF under grant CCF-2005804, USDA/NIFA under grant USDA-NIFA:2021-67021-35329, and ARPA-E under grant DE:AR0001215.

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

# Differentiable Dynamic Programming

Dynamic Programming (DP) is a class of methods that can efficiently solve many combinatorial optimization problems that would otherwise not be solvable in polynomial time. DP algorithms exploit the existence of overlapping sub-problems by storing the solutions to these sub-problems for later use (Bellman 1957). All DP approaches can be formulated as operations on Directed Acyclic Graphs (DAGs), with topologically ordered nodes $v_1, .., v_n$ and edge weights $\theta \in \mathbb{R}^{N \times N}$ (In the case where each node's degree is bounded by some constant $D$, we equivalently have $\theta \in \mathbb{R}^{N \times D}$). The problem then reduces to finding the optimal path along with the graph, which maximizes (or minimizes) the sum of edge weights. This can be done through the following recursive computation:

$$v_1(\theta) \triangleq 0 \tag{7}$$
$$\forall i \in [2, ..., N] : v_i(\theta) \triangleq \max_{j \in \mathcal{P}_i} \theta_{ij} + v_j(\theta) \tag{8}$$

The final output $V(\theta) \triangleq v_N(\theta)$ is the optimal cost and can be shown to be the same as a result obtained by performing an exhaustive search over all possible paths. By storing the maximizing index for each step, the *solution* of the dynamic program can then be reconstructed through backtracking from the final node of the graph, giving the optimal path.

Although this mapping $V(\theta) : \mathbb{R}^{N \times N} \to \mathbb{R}$ is non-differentiable wherever the solution is not unique, previous work (Mensch and Blondel 2018) has devised a method for computing a differentiable approximation by adding a strongly convex regularizer to each of the recursive updates. For an arbitrary regularizer $\Omega$, they leverage the smoothed max operator $\max_{\Omega}(x)$ introduced by (Nesterov 2005), which is defined as follows:

$$\max_{\Omega}(x) = \max_{q \in \Delta^{|x|}} \langle q, x \rangle + \Omega(q) \tag{9}$$

Unlike the regular max operator, this smoothed version is continuously differentiable everywhere, and by substituting it into the recursive updates in Equation 2, we can define a smooth DP approximator $V_{\Omega}(\theta)$ with gradients $\nabla_{\theta} V_{\Omega}(\theta)$ that are guaranteed to exist. The approximation is also principled in the sense that $\lim_{\alpha \to 0} V_{\alpha\Omega}(\theta) = V(\theta)$ for any choice of $\Omega$. In this work we focus on using negative entropy as the regularizer $\Omega$, which yields the well-known softmax function (Bridle 1990). However, (Mensch and Blondel 2018) also shows $L_2$ regularization to be useful for encouraging sparsity. For DP problems that call for minimization over arguments, it is straightforward to define an analogous smoothed-min operator $\min_{\Omega}(x) \triangleq -\max_{\Omega}(-x)$.

To allow the computation of input gradients $\nabla_{\theta} V_{\Omega}(x)$ in the backward pass, the maximizing $q$ vectors from (3) must be stored in the forward pass, with $q_i$ denoting the maximizing argument used to compute $v_i$. Together, these define transition probabilities for a random walk on the input graph, starting from the final node and working backwards. The backward pass consists of backtracking through the DAG to compute the marginal probability of each node and edge being visited during this random walk. The marginal proba-

bility of each edge being visited is equivalent to the gradient of $V_{\Omega}(\theta)$ with respect to that edge.

In addition to this methodology for relaxing the cost of the DP, (Mensch and Blondel 2018) also show that the solution of the dynamic program can be relaxed and that its gradient with respect to the input is equivalent to the Hessian matrix of $V_{\Omega}(\theta)$. Rather than explicitly computing the Hessian, however, they use a variation of Pearlmutter's method (Pearlmutter 1994) to only compute the product of the Hessian with a specific matrix to get the derivative in that direction.

# Game Design

We first compare the design capabilities of backward-pass approximation to an optimal forward-pass approximation on a simple game called Monster Trainer, first introduced in (Xu et al. 2020). The input is a set of monsters, each represented by a 2-dimensional location $h \in [0, 10]^2$ and a $c \in [1, 10]$. Starting at a random location with level 0, the trainer can advance to level $n$ by challenging the monster with combat level $n$, but incurs a cost equal to the distance from their current location to the monster multiplied by their difference in levels. After challenging the monster, the trainer then moves to its location. The game ends when the trainer reaches the target level provided to them at the beginning of the game.

The minimum cost needed to complete a level of Monster Trainer can be found using the Bellman-Ford algorithm for shortest paths (Bellman 1958), a DP computation. However, (Xu et al. 2020) also show that Graph Neural Networks (GNNs) are extremely well-suited to solve this problem, given sufficient input and output examples. They argue that because the GNN computational structure resembles that of a DP algorithm, GNNs can easily generalize from examples to learn the DP function. As evidence of this, they are able to train a GNN on randomly generated levels of the Monster Trainer game, achieving minimal validation loss (MSE of 0.564 on test set).

Because GNNs are differentiable by design, they admit an input gradient which can be easily computed using back-propagation alone without any approximation in the backward pass. In our experiment, we measure the usefulness of this GNN gradient compared to a data-free approximation of Bellman-Ford. This makes for a compelling comparison as the GNN has learned the update rules of Bellman-Ford and the differentiable DP method has directly relaxed the very same updates through entropy regularization.

For each gradient approximation, we test its ability to create Monster Trainer levels whose minimum cost to completion meets a designers' specifications. To do this, we begin with a random game level and shift each monsters' coordinates in accordance with the gradient of the shortest path. 6a shows the effect of using gradient ascent to increase the shortest path values and 6b shows the results of using gradient descent to decrease them.

Our results show that gradients obtained using backward-pass approximation allow for much more stable adjustment of the shortest-path attribute, especially at higher learning

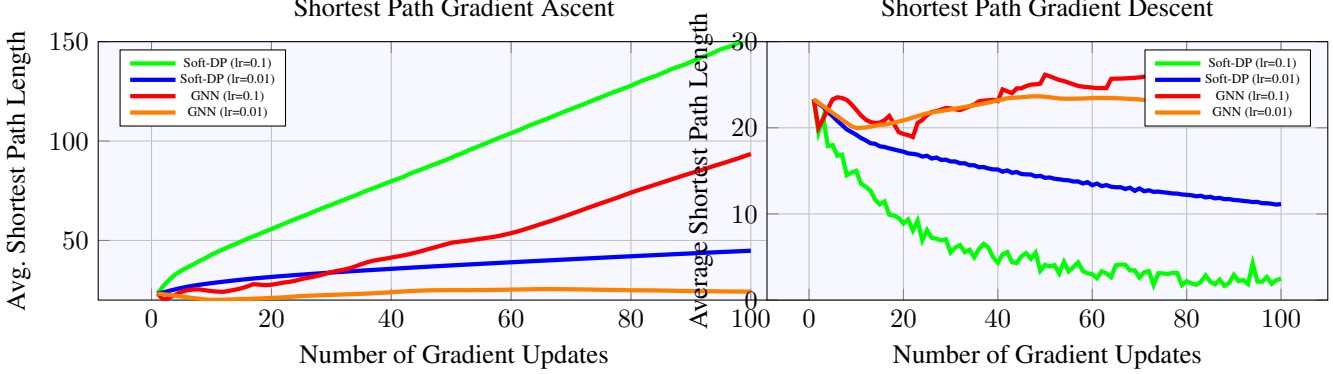

(a) Gradient Ascent on Monster Game Levels

(b) Gradient Descent on Monster Game Levels

rates. In many cases, the forward-pass approximation obtained from the GNN shifts the shortest-path value in the opposite direction from the one intended, despite its tight approximation of the DP-function and exact gradient computation.

### Optimal Histogram Differentiation

We consider the following toy problem. Let $\mathbf{x} \in \mathbb{R}^n$ and $\mathbf{y} \in \mathbb{R}^n$ be an input and $k$-*histogram* signal, respectively. Since solving $\min_{\mathbf{x}} \|\mathbf{x} - \mathbf{y}\|_2^2$ based on descent method is a trivial, we consider the optimization problem with a noisy target $\hat{\mathbf{y}} = \mathbf{y} + \epsilon$ where $\epsilon \sim N(0, \sigma^2)$. Given a prior that the target signal $\mathbf{y}$ is $k$-*histogram*, we solve a following optimization problem:

$$\min_{\mathbf{x}} \|h(\mathbf{x}) - \hat{\mathbf{y}}\|_2^2 \qquad (10)$$

and measure the distance between histogram approximation of updated input $h(\mathbf{x})$ and clean target signal $\mathbf{y}$.

We compare $h$ with following options: Soft-DP, DSA, and surrogate models based on neural networks. We prepare pretrained multi-layer perceptron (MLP) and convolution neural networks (CNN) to solve $k$-*histogram* approximations as a surrogate model using 100K randomly partitioned $k$-*histogram* with $k=3$. We freeze the surrogate models' parameters when solving the optimization problem in Equation 6. We define the input $\mathbf{x}$ to be noisy $k$-*histogram* signals which in range of $[0, 100]$ and $\mathbf{y}$ to be clean $k$-*histogram* generated independent to $\mathbf{x}$ (equivalently, partitions and values independent). Given the choice of $h$, we update the input $\mathbf{x}$ $2,000$ iterations with learning rate 10 via SGD method. We choose the $\sigma = 5$ for $\hat{\mathbf{y}}$ to be the same value to train the surrogate models. Our soft-dp approximation provides the most consistent approximation to the clean target as shown in Figure 2.

We also compare the Jacobian $\partial h / \partial \mathbf{x}$ between Soft-DP, DSA, MLP, and CNN given histogram inputs $\mathbf{x}$ as shown in Figure 3. DSA jacobian is the exact Jacobian given the histogram approximation. Soft-DP adds the probabilistic aspect to Jacobian that neighboring block matrix smooths out if the truth signal partitions are close in terms of the value. On the other hand, we observe MLP and CNN having some noise to the Jacobian.

### Microstructure Design

In materials science, it is common to represent materials by their 2-dimensional microstructures which display the arrangement of their components. These microstructures determine the properties of the materials to which they correspond. Designing microstructures which satisfy key constraints therefore allows for the controllable synthesis of novel materials.

Previous work from (Joshi et al. 2020) has introduced the InvNet model, which augments the loss function and training procedure of a standard Wasserstein-GAN (Arjovsky, Chintala, and Bottou 2017) to train a controllable generator. In addition to the standard noise vector input $z$, the InvNet generator also accepts a control vector $c$ which represents the desired value of some attribute of the data sample that can be computed by some function $f$. The standard WGAN loss function (11), can then be replaced with $\overline{L}$ (12)

$$\mathcal{L}(\theta, \psi) = \mathbb{E}_{x \sim p_{\text{data}}} [D_\psi(x)] - \mathbb{E}_{x \sim p_z, c \sim p_c} [G_\theta(D_\psi(z, c))] \qquad (11)$$

$$\overline{\mathcal{L}}(\theta, \psi) = \mathcal{L}(\theta, \psi) + \mathbb{E}_{x \sim p_z, c \sim p_c} \left[ \lambda_I (c - f(G_\theta(z)))^2 \right] \qquad (12)$$

InvNet also uses a unique three-way optimization method to find the Nash equilibrium of the min-max optimization problem $\min_\theta \max_\psi \overline{L}$ Note that for this mini-max optimization to be tractable, the attribute function $f$ must be differentiable. The applications of this model in prior work include the microstructure synthesis problem mentioned above, in which the attribute functions $f$ were chosen to be the volume fraction and two-point correlation which both have easily defined gradients.

One important attribute to control in the design of materials used in photovoltaic applications is conductivity, which corresponds to the shortest path from the top of the array to the bottom along a single material. Motivated by this, we trained an InvNet model on a dataset of 2-phase microstructures and selected an invariance function $f$ defined as the shortest-path from the top to the bottom of the image under some pairwise distance function $d$, which measures the difference between adjacent pixel values. Note that in the following definition, we ensure that $p_1$ is in the first row of the

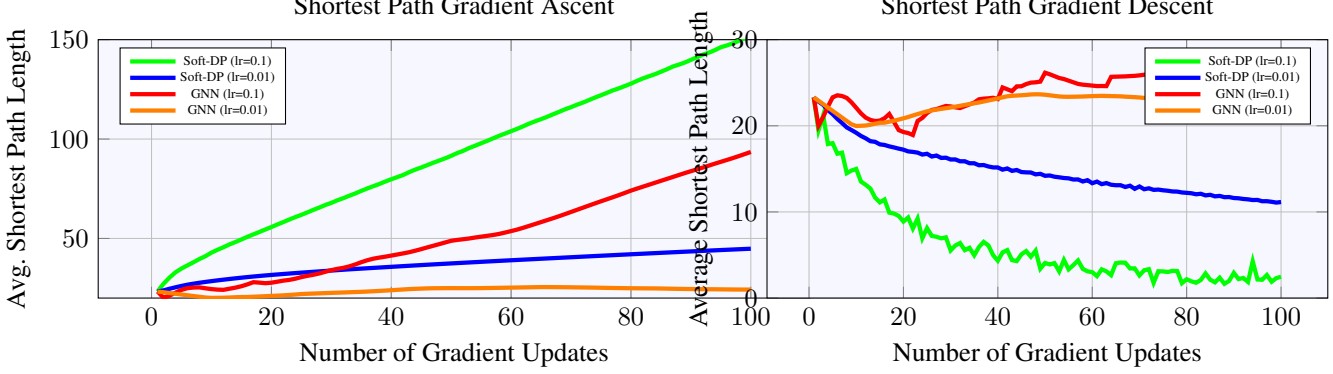

(a) Gradient Ascent on Monster Game Levels

(b) Gradient Descent on Monster Game Levels

array, $p_2$ is in the last row, and that $\forall i : p_i, p_{i+1}$ are adjacent to one another

$$SP(\mathcal{I}) = \min_{p_1, p_2 \ldots, p_h} \sum_{i=1}^{h-1} d(\mathcal{I}_{p_i}, \mathcal{I}_{p_{i+1}})$$

To ensure that identical pixels would have a distance of 1 and that the distance increases rapidly as the values diverged, we chose our distance function as $d(x_1, x_2) = \exp((x_1, x_2)^2)$.

To use $SP$ as our attribute function, we first formulate it as a straightforward DP problem, with its cost representing the length of the shortest path. We then added entropy regularization to approximate $SP$ with a differentiable function $SP_\Omega$. Due to the high number of DP updates needed to compute $SP_\Omega$ (Each microstructure was of size $64 \times 64$, making for a total of $2^{12}$ DP updates), we found it necessary to use our proposed debiased soft-DP approach to ensure accurate gradients. This setting gives us an opportunity to test whether the soft-DP gradients can be used to train deep neural networks in addition to the simple input updates we've seen previously.

Our experiments show that the shortest-path relaxation is effective in reducing the invariance cost $(c - SP(x))^2$, meaning that the generator is able to learn to output samples with specified shortest-path values. Increasing the penalty coefficient for the DP-based loss further ensures that the generator obeys the control input.

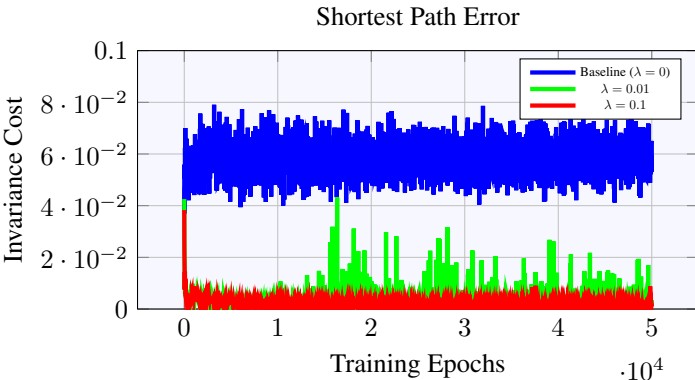

Figure 7: Projection Error

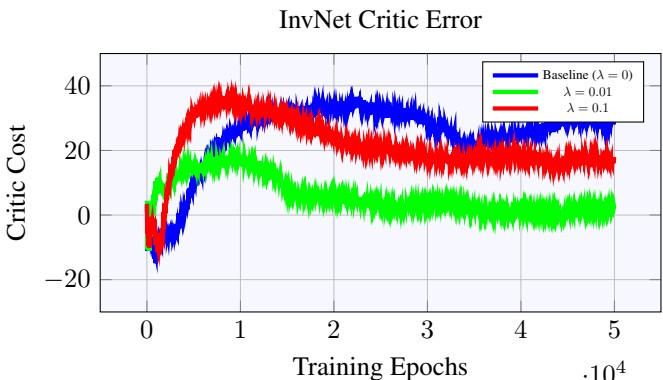

Figure 8: Critic Error