# OpenReview forum: "Differentiable Design With Dynamic Programming"
_AAAI.org/2022/Workshop/ADAM — AAAI 2022 Workshop ADAM_

### Official Review · Reviewer_cmVr · 2021-11-30
**Good work; minor suggestions**

**Rating:** 7
**Confidence:** 4

**Review:**

In this paper, the authors propose a *soft-DP* framework which is a differentiable (i.e., can be used in gradient-based learning systems such as deep learning) dynamic programming framework. This framework has several advantages as pointed out by the authors including tasks such as histogram-approximation, and material microstructure reconstruction. The idea is novel and relevant to this workshop with large applicability in design and manufacturing problems.  The results are comparable with some of the state-of-the-art methods (DSA) and also traditional methods with just an MLP or a CNN.

The following are my minor comments:
1. There is some typo near Equation 4 making it unclear. Further, it is not clear how inaccuracy of $V_{\Omega}$ will prevent $V(x)$ from converging to $c$. Isn't this the regular chain rule and regular MSE loss used all along in all training protocols? The authors could be a bit more specific on what is not allowing the  $V_{\Omega}$ to converge to $c$.
2. The authors introduce the term debiased soft-DP but do not show the proper comparison between soft-DP and debiased soft-DP for a given problem.
3. A general note is while the authors claim that soft-DP (or for that matter any differentiable programming approach) is much economical than training a surrogate, there is no proof to claim this. Specifically, since in each forward pass of the debiased soft-DP, actual $V$ is calculated to reduce the bias. It seems that training a surrogate may give better results.
4. While this paper is good for this workshop, more quantitative comparisons and ablations studies may be needed for improving this work.

---

### Official Review · Reviewer_CmNy · 2021-12-01
**The authors have proposed a differentiable design technique for dynamic programming, called soft-DP. This has been demonstrated for target matching problems, and the framework has been validated for three applications - game design, histogram approximation and materials design.**

**Rating:** 9
**Confidence:** 1

**Review:**

The work rates high on originality, quality and significance. A strong point of this work is identifying the advantage of soft-DP over data-heavy approaches. If possible within the scope of this paper, some clarity is desired on the following points:

1) In Fig. 2, MLP and CNN have been reported to have overfitted the noisy targets. Could you elaborate on the MLP/CNN architectures? Can the overfitting be improved with hyperparameter tuning of the surrogates?
2) Any thought on adapting this methodology for situations when the DP updates are expensive? For example the microstructure design problem involves 2^12 DP updates, which can grow exponentially when the problem becomes more complex with more than two phases. How do we tractably adapt these for situations when the underlying DP updates are expensive to compute?


Some minor typographical errors to be corrected:
1) Page 2. Before equation 4: To get a differentiable loss, we first V with $V_\sigma$. - Please check for correctness
2) Page 2.  "The key lies in observing that even when we cannott" - spelling of can not
3) Page 2. Gradients of solutions. "In several problem" - should be "problems"
4) Page 2. "which maps an signal" - should be a signal